

# Life as a bachelor: quantifying the success of an alternative reproductive tactic in male blue monkeys

Su-Jen Roberts[1,2,3] and Marina Cords[1,2]

[1] Department of Ecology, Evolution, and Environmental Biology, Columbia University, New York, NY, USA
[2] New York Consortium in Evolutionary Primatology, New York, NY, USA
[3] New Knowledge Organization Ltd., New York, NY, USA

Corresponding author
Su-Jen Roberts,
sroberts@newknowledge.org

## ABSTRACT

In species that live in one-male groups, resident males monopolize access to a group of females and are assumed to have higher reproductive success than bachelors. We tested this assumption using genetic, demographic, and behavioral data from 8 groups of wild blue monkeys observed over 10 years to quantify reproduction by residents and bachelors and compare the success of the two tactics. We used maximum-likelihood methods to assign sires to 104 offspring born in the study groups, 36 of which were sired by extra-group males, i.e., residents of neighboring groups and bachelors. Among these extra-group males, high-ranking males (many of whom were neighboring residents) were more likely to sire offspring than low-ranking males, but the time these visiting males spent in the mother's group when she conceived (male presence) did not predict their relative success. When bachelors competed for reproduction with other bachelors, neither rank nor male presence during the mother's conceptive period affected the probability of siring an offspring, suggesting that highly opportunistic mating with conceptive females is important in bachelor reproduction. In a second analysis, we used long-term data to estimate resident and bachelor reproductive success over the long term, and particularly to determine if there are any circumstances in which a typical bachelor may sire as many offspring as a typical resident during one or two periods of residency. Our findings generally support the assumption of a resident reproductive advantage because in most circumstances, a lifelong bachelor would be unable to sire as many offspring as a resident. However, a bachelor who performs at the average rate in the average number of groups for several years may have similar lifetime reproductive success as a male whose reproduction is limited to one short period of residency, especially in a small group. Our findings suggest that one should not assume a resident reproductive advantage for males in one-male groups in all circumstances.

## INTRODUCTION

In species with strong male–male competition for females, individual males often adopt alternative reproductive tactics to maximize reproductive output within the constraints of social, demographic, and ecological conditions (*Gross, 1996*; *Neff & Svensson, 2013*). In male mammals, alternative reproductive tactics usually have unequal fitness payoffs; the most competitive males use the tactic with the highest payoff, while less competitive males use surreptitious behavior to gain access to females, resulting in lower reproductive success (*Wolff, 2008*; *Taborsky & Brockmann, 2010*). Individuals may, however, switch tactics over the course of a lifetime or even during a single breeding season, in response to factors like their relative competitive ability and the proportion of males pursuing each tactic (*Repka & Gross, 1995*; *Taborsky & Brockmann, 2010*).

Many mammals live in one-male/multi-female groups, a type of social organization in which one resident male is consistently associated with a group of females while bachelor males live alone or form bachelor groups. Males compete for the resident position (e.g., *Le Boeuf, 1974*; *Clutton-Brock, 1982*), suggesting that the resident tactic of defending access to a group of females results in higher fitness than the bachelor tactic of stealing matings. Paternity studies from several mammals living in one-male groups strongly support this prediction, in that residents sired all offspring born in their groups (*Schwartz & Armitage, 1980*; *Pope, 1990*; *Launhardt et al., 2001*), which suggests that lifelong bachelors do not sire any offspring. In other mammals, however, residents lost some paternity to outsiders, including bachelors and residents of adjacent groups (*Pemberton et al., 1992*; *Asa, 1999*; *Hoelzel et al., 1999*; *Storz, Bhat & Kunz, 2001*; *Heckel & Von Helversen, 2003*; *Fabiani et al., 2004*; *Feh & Munkhtuya, 2008*; *Hirsch & Maldonado, 2011*; *Roberts, Nikitopoulos & Cords, 2014*). While an alternative reproductive tactic may exist in these species, the success of individual bachelors and the factors affecting whether they sire offspring are not well understood. Furthermore, few studies have directly compared the success of bachelor and resident tactics in long-lived mammals (but see: *Sommer & Rajpurohit, 1989*; *MacLeod, Ross & Lawes, 2002*) because it is difficult to follow males over their entire reproductive lives. Quantifying the success of alternative tactics and identifying the conditions under which each does best allows us to better understand the role of male–male competition in determining the evolution of social organization.

We studied male reproductive tactics in blue monkeys (*Cercopithecus mitis*), a guenon that lives typically in one-male groups. Even as a long-term resident, male blue monkeys regularly face competition from extra-group males, who may be bachelors, unattached to any group, or residents in adjacent groups (n.b., there are no all-male groups). These extra-group males compete with a resident by spending time in his group during the mating season, and copulating with his females (*Cords, 2000*). Neighboring residents are especially likely to steal copulations during aggressive intergroup encounters (which involve primarily the females; *Cords, 2007*). In addition, about 25% of mating seasons in our study population are characterized by particularly strong male–male competition when multiple extra-group males temporarily join the group for various periods (from days to months), leaving when the mating season ends (*Cords, 2002*). These multi-male

influxes are especially likely when multiple females are simultaneously sexually active (*Cords, 2002*; *Mugatha et al., 2007*) and they reduce the probability that a resident male sires offspring conceived in his group (*Roberts, Nikitopoulos & Cords, 2014*). These findings suggest that stealing matings may be a profitable bachelor tactic (*Rowell & Chism, 1986*).

Individual bachelors in the study population typically sire fewer offspring than residents within a single group-year (*Roberts, Nikitopoulos & Cords, 2014*), but there is variation in bachelor success, and its causes are as yet unexplored. In addition, little is known about how resident or bachelor tactics translate into lifetime reproductive success. Comparing resident and bachelor reproductive success would help determine if the high costs associated with residency allow bachelors to make up some of the difference in reproduction by pursuing the lower-cost, lower-gain tactic for an extended period (*Widemo, 1998*; *MacLeod, Ross & Lawes, 2002*). In this study, we used molecular, demographic, and behavioral data collected from 8 social groups of wild blue monkeys over 10 years to assess the absolute and relative success of the bachelor tactic. We present our methods and results in two parts to increase clarity: Part I focuses on the factors that drive differential siring success among extra-group males, while Part II compares the lifetime reproductive success of bachelors vs. residents.

## PART I: FACTORS AFFECTING EXTRA-GROUP MALE SIRING SUCCESS

### Part I Methods

#### Ethical note

This research adhered to the Animal Behavior Society guidelines for the treatment of animals in behavioral research. Methods were approved by the Columbia University IACUC (# AC-AAAD9003), the National Council for Science and Technology, the Kenya Wildlife Service and National Environmental Management Authority.

#### Study site and population

The study population of blue monkeys in the Kakamega Forest, Kenya (*Mitchell, Schaab & Wagele, 2009*) has been monitored since 1979 (*Cords, 2012*). Our study focused on the period between 2002 and 2011, during which 3 group fissions increased the number of simultaneously existing study groups from 3 to 6. We used data from a total of 8 unique groups, each present for 1–10 years during the study period. Authors and field assistants observed groups on a near daily basis to record demographic information including infant birthdates, presence of sexually active females and of extra-group males (both neighboring residents and bachelors) in or on the edge of the group, and dates of resident male turnover. They also recorded all observed agonistic interactions among males, noting participant identity and outcome.

#### Genetic data

We used fecal samples from 126 infants, 64 mothers, and 60 adult males, including 11 resident males in our study groups, for genetic analysis. We collected samples shortly

after defecation, storing them in sterile tubes mixed with RNALater™(Ambion) to preserve the DNA.

Extraction and genotyping methods are described in *Roberts, Nikitopoulos & Cords (2014)*. Briefly, we extracted DNA from fecal samples using the QIAamp® DNA Stool Mini Kit (Qiagen, Hilden, Germany) and amplified it at 13 human MapPairs® microsatellite markers (Invitrogen, Carlsbad, California, USA). We used the ABI 3730 Automated DNA Analysis system and *GeneMapper 3.7* (Applied Biosystems, Carlsbad, California, USA) for genotyping.

We conducted likelihood-based paternity analysis with Cervus *3.0* (*Kalinowski, Taper & Marshall, 2007*), using simulation parameters noted in *Roberts, Nikitopoulos & Cords (2014)*. Cervus assigned paternity to 108 offspring with 80% or 95% confidence (for details, see *Roberts, Nikitopoulos & Cords, 2014*). We further verified assignments by examining the number of mismatches between the offspring and assigned sire, excluding mismatches for which both individuals were homozygous because they may have resulted from allelic dropout. Mother–offspring pairs mismatched rarely and never at more than 1 locus. Similarly, 104 of the 108 offspring-assigned sire pairs mismatched at 0 or 1 locus and we considered these assignments to be robust. The remaining 4 offspring mismatched their assigned sire at 2 loci, so we omitted them from the final analysis.

### Behavioral data

We used demographic data to identify factors affecting the siring success of extra-group males. We were interested in how extra-group males compete among themselves for reproductive opportunities so considered only the offspring sired by bachelors or residents of adjacent groups (i.e., neighboring residents). Specifically, we tested how male presence and dominance rank affected the probability of siring an offspring.

Blue monkey females do not signal fertile periods with morphological changes such as sexual swellings, so we identified the period in which an infant's conception occurred (conceptive period) using a combination of demographic records and behavioral observations. First we identified a 29-day conception window for each offspring by subtracting the length of one gestation ($176 \pm 14$ days, the 95% confidence interval, *Pazol, Carlson & Ziegler, 2002*) from offspring birthdates, which were accurate to 1–3 days. Within this window, we then identified the conceptive period based on female sexual behavior (copulations and proceptive behavior), following *Pazol (2003)* and *Roberts, Nikitopoulos & Cords (2014)*. We identified conceptive periods for the mothers of 28 of the 36 offspring sired by extra-group males. The mean length of these periods was $5.4 \pm 5.1$ days (range = 1–22 days, $N = 28$).

We summarized our data as offspring–male pairs, each corresponding to one male observed in or near the group during the conceptive period of the mother of each offspring. We included offspring–male pairs only if the male was individually identified and genotyped because we were unable to predict siring success for males who we could not readily identify or for whom we lacked genetic data. Individually-identified and genotyped males comprised most of the males observed in the groups during conceptive periods (mean $\pm$ sd = $83 \pm 18\%$, range = 44–100%, $N = 28$ offspring). Our data sets

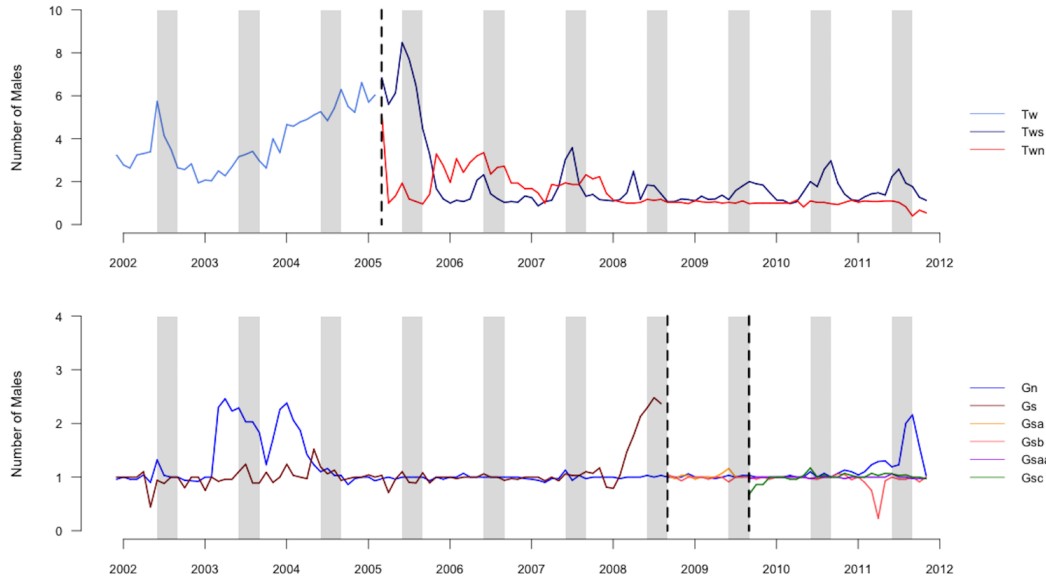

**Figure 1 Graph of number of males in groups.** Average number of males (including the resident male) present in each group averaged by month during the study period. Vertical dashed lines indicate dates of group fission events. Gray columns indicate the population peak in conceptions, which occurs from July until September.

did not include the resident of the group in which the offspring was conceived because we focused on competition among extra-group males. The average number of males present in a group per month varied by group and time of year, with peaks typically occurring during mating seasons (Fig. 1; *Cords, 2000*). The mean ± SD number of extra-group males present per conceptive period was 2.5 ± 2.4, which included 2.2 ± 2.1 bachelor males.

### *Factors determining siring success of extra-group males*

Our first predictor, male presence, was the proportion of days in the mother's conceptive period in which we observed the male in or on the edge of her group. Our second predictor was male dominance rank. Although male blue monkeys do not regularly live in groups together, they do interact agonistically, thus establishing dominance relationships that may order them into a queue for reproduction (*Alberts, Watts & Altmann, 2003*). We assigned ranks using the Elo-rating method (*Albers & de Vries, 2001*; *Neumann et al., 2011*) implemented with the EloRating package in R (version 0.98.1102). In this method, each individual is assigned an initial Elo-rating, which is recalculated after each dyadic contest, with the winner gaining and the loser losing points. The number of points gained or lost is based on the expected probability of each participant winning the interaction. For example, if a higher-ranking male wins a contest against a lower-ranking male, his Elo-rating increases (and the lower-ranking male's Elo-rating decreases) only slightly. If, however, a lower-ranking male wins an interaction against a higher-ranking male, each male gains or loses more points. This method is particularly appropriate for our study system because it accommodates a varying number and identity of individuals over time, and allows an assessment of rank at any specific timepoint (*Neumann et al., 2011*).

Additionally, the Elo-method can use interactions between all males (including residents) to assign ratings, so bachelors can be ranked relative to each other even if they have not directly interacted.

We compiled the 2002–2011 records of dyadic agonistic behavior (chase, attack, lunge, bite, hit, avoid, flee, nasal scream) in which there was a clear winner and loser. All males had a starting rank of 1,000 and interacted for 6 months before we assessed their Elo-rating for our data analyses. For our analyses, we used a male's Elo-rating on the day representing the midpoint of the conceptive period for each offspring.

### Statistical analysis

We used conditional logistic regression (in Stata 13.1) stratified by infant to evaluate how male presence and rank affected the log odds of siring an offspring. We assessed the significance of entire models and individual predictors using likelihood ratio tests to compare models with the predictor(s) present versus absent. Conditional logistic regression requires variation in the dependent variable within each stratum, so for each offspring sired by an extra-group male, we had to observe both the sire and at least one non-sire (also an extra-group male) at conception. We therefore excluded any offspring for which we observed only one extra-group male at conception. Our first analysis included both neighboring residents and bachelors (both as sires and non-sires). We also assessed how rank and male presence affected the relative success of bachelor males only by generating a second data set that included only bachelor males (both as sires and non-sires).

## Part I Results

Of the 104 offspring with verified paternity assignments, 68 were assigned to residents. The remaining 36 were assigned to extra-group males, 12 to residents of adjacent groups and 24 to bachelors.

### Factors determining siring success of extra-group males

When we considered offspring sired by any extra-group male, a conditional logistic model including both male presence and rank was better than a null model at predicting which male sired an offspring (likelihood ratio test: $\chi^2 = 13.81$, $P = 0.001$, $N = 11$ offspring with one sire and 1–7 non-sires each). However, the effect of male presence was not significant (Wald $P = 0.943$ for rank model), and likelihood ratio tests showed that a model including this predictor did not differ from one that excluded it (rank: $\chi^2 = 0.01$, $P = 0.9433$). We concluded that male presence was not a useful predictor of siring probability among extra-group males, while rank had significant effects. A final univariate model showed that an extra-group male increased his odds of siring an offspring by a factor of 1.01 with a one-step increase in Elo rating (odds ratio = 1.007, Wald $P = 0.008$; likelihood ratio test, $\chi^2 = 13.809$, $P = 0.0002$). The average number of Elo-steps between two adjacently ranked males was 154, so the odds of siring an offspring would increase by a factor of approximately 156 for a one-step increase in ordinal rank. This large rank advantage occurred because the highest-ranked male sired eight of the eleven offspring in our data set and in only one case did a male more than 79 Elo-steps below the highest-ranked male

sire the offspring. Overall, the average Elo rating of sires was 1,181 ($N = 11$), while for non-sires it was 902 ($N = 35$).

We analyzed the effect of male rank and male presence on the log odds of siring an offspring for a subset of the above cases in which bachelors were the sires, and here included only other bachelors as non-sires ($N = 6$ offspring, with one sire and 1–4 non-sires each). For bachelors competing amongst themselves, neither male presence (Wald $P = 0.645$) nor rank (Wald $P = 0.180$) predicted siring an offspring, and a model including these predictors was not significantly different from a null model (likelihood ratio test, $\chi^2 = 3.031, P = 0.220$).

Because conditional logistic regression requires both positive and negative outcomes (sires and non-sires for each offspring), our analyses could not use data from (i) 7 infants who were known *not* to be the offspring of their group's resident male, even though they were not assigned to any other male that was present in their group at conception, or (ii) 3 infants whose paternity was assigned to the single extra-group male that visited the group at the time of their conception. These cases can provide information about both non-sires and sires, but one cannot stratify the comparison by offspring. When we considered both neighboring residents and bachelors as sires and non-sires for all 21 offspring, we again found that sires had significantly higher Elo-ratings (median for sire: 1,222 vs. non-sires: 912; Mann Whitney U Test, $U = 649.5, Z = -3.77$, 2-tailed $P = 0.0002, N = 14$ sires, 56 non-sires) but there was no difference for male presence ($U = 357.5, Z = 0.50, P = 0.6171$). When we considered only those offspring sired by bachelors ($N = 8$), we found no significant difference in rank or male presence between bachelor sires and non-sires (rank: $U = 68, P > 0.05$; time in group: $U = 63, P > 0.05, N = 8$ sires, 14 non-sires for both tests). Thus, these results incorporating additional offspring corroborated those from the conditional logistic regression.

## PART II: COMPARING RESIDENT AND BACHELOR SUCCESS

### Part II Methods
#### *Comparing resident and bachelor tactics*
Comparing the long-term success of resident and bachelor tactics required estimating reproduction across multiple groups and years. While our paternity data spanned multiple groups and years, some parameters were not directly measured and were inferred based on known patterns of reproduction in study groups. As such, our calculations were exploratory and served to determine if there are any conditions under which bachelorhood and residency may be comparably successful reproductive tactics.

Our calculations were based on *MacLeod, Ross & Lawes (2002)* who aimed to discover if, over an entire reproductive lifetime, a bachelor samango monkey could steal enough matings to be as successful as a resident male siring at the average rate for one average period of tenure. These authors concluded that a bachelor would have to pursue the steal strategy for 15.1 years to obtain the same number of matings as a resident in one average period of tenure. They judged 15 years to exceed the length of the male reproductive

lifespan and concluded that a resident male with just one period of tenure would always obtain more matings than a bachelor. Importantly for our purposes, however, this calculation made four assumptions: (1) the resident confines his reproduction to his group, (2) the resident loses reproduction to only one bachelor, (3) the bachelor confines his reproduction to one group and (4) that matings map directly onto reproduction. Our paternity results indicated that our study population violated all four assumptions, so we developed a new calculation for comparing bachelor and resident siring success to account for siring success in multiple groups and by multiple bachelors within one group. Specifically, we identified components of bachelor and resident reproduction and used summary statistics to determine how long the average bachelor would have to pursue the bachelor tactic to match the siring success of the average resident.

As acknowledged earlier, blue monkey males may switch among tactics during their lives, so some males may sire offspring as bachelors before or after attaining residency. Genetic and behavioral evidence suggest, however, that a resident's siring success is concentrated during residency. In the study population, males who became residents appeared to gain this status around the time they attained full adult body size. As we almost never observed young adult males (i.e., those that are not full grown) copulating, the window in which males that become residents are old enough to copulate but not yet resident in a group is very short, thereby limiting pre-residency siring success. Supporting this inference, only 1 of 8 males sired offspring in a study group before attaining residency and all these offspring were in the group with an infertile resident. We know less about post-residency siring success because many deposed residents (e.g., 7 of 18 during the study period) disappear. However, demographic records indicate that 5 of the 11 (45%) residents with known fate died and genetic data indicate the other six did not sire offspring as a post-resident bachelor. We therefore computed a resident's siring success based only on his period of residency.

Additionally, 4 of the 11 resident males in our study groups were known to have two periods of residency (in different groups). We therefore computed a resident's siring success based on one or two periods of residency.

### Resident parameters

Resident siring success during his tenure is expressed as follows (Eq. (1)):

$$R = \sum_{t=1}^{T} (R_i + R_o N_r)_t \tag{1}$$

$R_i$ is the annual siring rate (number of offspring) in the resident's own group, $R_o$ is his annual siring rate per outside group, $N_r$ is the number of outside groups, and $T$ is tenure length, measured as the number of mating seasons (one per year in blue monkeys). The product of $R_o$ and $N_r$ equals "outside-group" siring success, so the value in parentheses is equal to the resident's total annual siring success, which is summed over his entire tenure.

We estimated within-group siring success ($R_i$) by summing the number of offspring sired in a resident's group during his entire tenure and dividing by the number of years he
was resident. One resident, Ro, did not sire any of the 19 offspring conceived in his group despite mating during most conceptive windows, which suggests that he was infertile. We therefore calculated summary statistics for siring success excluding Ro.

There are likely more reproductive opportunities in larger groups, so we calculated $R_i$ for small and large groups separately. Small groups contained ≤15 females over age 5 (the age at which females begin to exhibit proceptive behavior) and large groups contained >15 females over age 5. Group fissions during the study period resulted in two study group residents (Sa and Pu) being represented twice, once as resident of the parent group and once as resident of one of the daughter groups.

Tenure length ($T$) equaled the number of mating seasons a resident male was in his group, using observations of 23 residents for which we observed complete periods of tenure since 1994. Most conceptions occur during the 5-month mating season from June–October (*Cords & Chowdhury, 2010*) and we considered a resident to be present for a mating season if he was in the group for more than half of this 5-month period.

We estimated the number of outside groups in which a resident may sire offspring ($N_r$) as the number of adjacent groups, because a resident has never been observed more than one home range away from his own group. Group members collaborate to defend territories against adjacent groups (*Cords, 2007*) and we used observations of intergroup encounters from 2002–2012 to assess the number of adjacent groups for each study group. Group fissions and home range shifts may cause the number of adjacent groups to change over time, so we assessed this number annually. In a few cases, the identity of an adjacent group was ambiguous because it was a non-study group without readily identifiable individuals in an area where multiple group home ranges overlapped. In those cases, we calculated the minimum and maximum number of adjacent groups and used the average of those values. Minimum values reflected the number of identified adjacent groups and maximum values included the unidentified adjacent groups, although it may have been a known group that was unrecognized at the time.

We estimated outside-group siring success ($R_o$) from resident paternity success in other study groups. We calculated this parameter for 20 residents with a total of 22 periods of residency by dividing the total number of offspring sired in adjacent study groups by the number of adjacent study groups and by tenure length.

### Bachelor parameters

Bachelor siring success is expressed as follows (Eq. (2)):

$$B = \sum_{l=1}^{L} (R_b N_b)_l \tag{2}$$

$R_b$ is the annual siring rate per group, $N_b$ is the number of groups encountered, and $L$ is a bachelor's reproductive lifespan. The value in parentheses equals the number of offspring a bachelor produces annually, which is summed over his entire lifetime.

We genotyped 47 males who were bachelors for part or all of the time they were observed in the study population and used long-term records to identify the years of
bachelorhood and study groups each male visited. To calculate annual siring rate per study group ($R_b$), we divided the total number of offspring a bachelor sired in the study groups by the total number of study groups and years that he was observed.

We later omitted some of these males from the calculations to ensure that the summary statistics were good estimates of bachelor parameters. First, we omitted 19 bachelors observed for only 1 year, as most were either subadults who had recently emigrated from their natal groups but had not yet left the local population or males that were passing through (i.e., seen only few times and could not be consistently identified). We omitted 3 bachelors who were not observed in any of the study groups because we were unable to assess their siring success. These 22 bachelors were particularly unlikely to sire offspring in the study groups (only one sired one offspring) and removing them increased bachelor siring rate. Of the remaining 25 bachelors, 10 eventually became residents in the study groups or adjacent non-study groups. To control for differences between "committed" bachelors and bachelors who later became residents, we limited our calculations to the 15 bachelors who were bachelors for the entire time we knew them, although we acknowledge that they may have been residents before or after they were observed.

We estimated the number of groups a bachelor encountered as the number of group home ranges that his home range overlapped. From June–September 2011, SJR walked transects following pre-existing linear footpaths spanning the study group home ranges. When she encountered a bachelor, she followed him and recorded his location every 30 min with a handheld GPS unit (Garmin GPSMAP® 60 CSx; Garmin, Schaffhausen, Switzerland) until she lost him. We have location data for 11 bachelor males, but used data from the 5 that were tracked for $\geq 15$ h each (mean $\pm$ sd $= 24.3 \pm 9.4$, range $= 16$–$38$ h over 5–11 days).

During the same period, we collected location data to map the home ranges of the 6 study groups. The field team contacted each group daily and recorded the location of the group center every 60 min with a handheld GPS unit. We recorded $414 \pm 167$ (mean $\pm$ sd) hours of location data per study group (range $= 139$–$579$) distributed over 81–123 days. SJR and one field assistant also tracked the location of 4 adjacent non-study groups, recording the location of the group center every 30 min for a mean of $74 \pm 30$ h of location data per group (range $= 37$–$110$) distributed over 14–19 days. We plotted all points on a $50 \times 50$ m grid. When a point fell within a grid cell, that cell was considered to be part of the group home range (Fig. 2).

We overlaid bachelor points on group home ranges and counted the number of group home ranges in which each bachelor was observed (i.e., the minimum number of groups encountered). Four of the 5 bachelors were observed outside the mapped group home ranges. Based on home range sizes of known groups and opportunistic observation of unidentified groups, we estimated the total number of group home ranges that bachelors may have overlapped (i.e., the maximum number of groups encountered). We used the midpoint of minimum and maximum values for each bachelor as the total number of groups encountered.

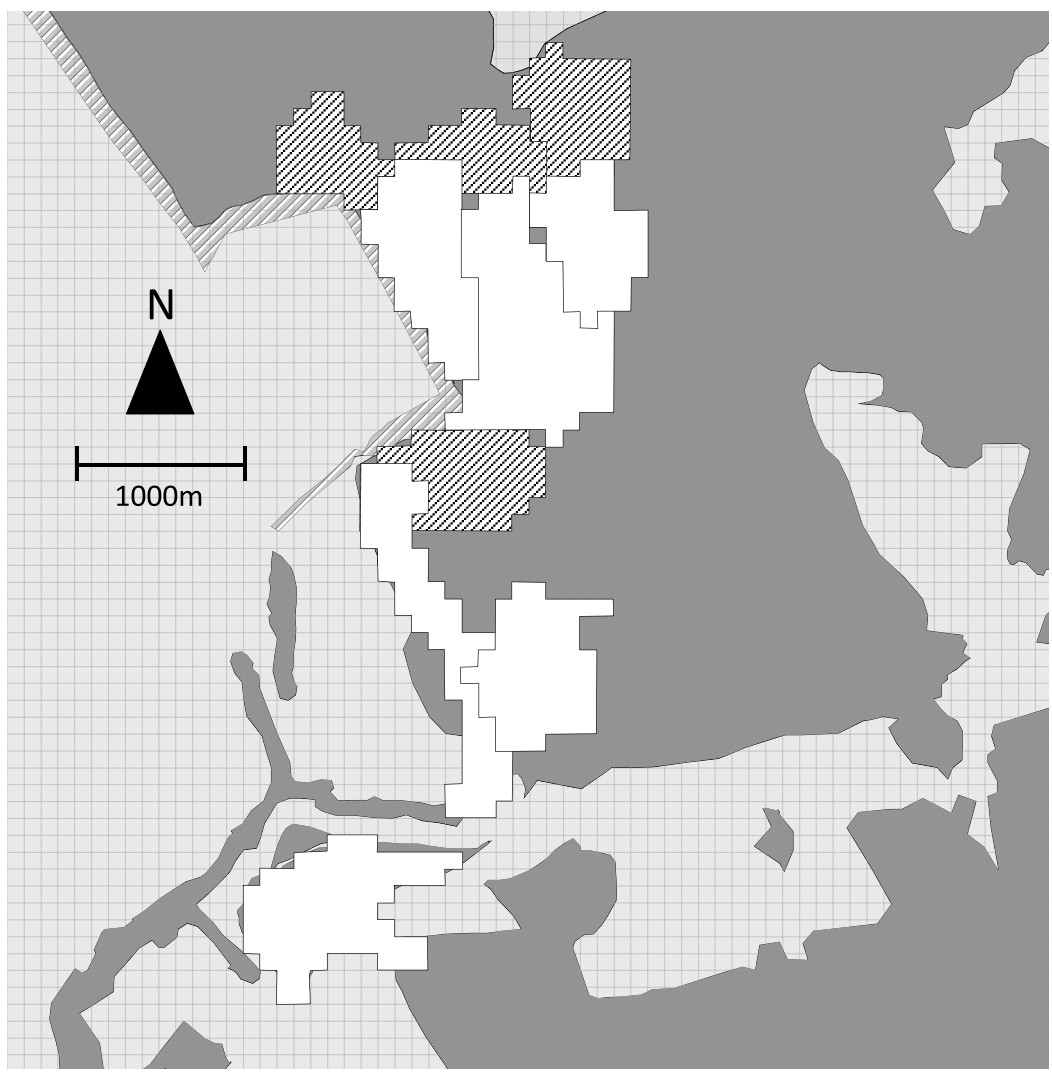

**Figure 2 Map of group home ranges.** Home ranges of 6 study groups (white) and 4 adjacent non-study groups (hatched). Dark grey indicates forested areas, light grey non-forested areas.

## *Comparing tactics*

We used mean values for each parameter to calculate the number of years a bachelor siring offspring at the average rate in the average number of groups (i.e., the average bachelor) would have to pursue the bachelor tactic to sire as many offspring as a resident siring at the average rate in his own group and at the average rate in the average number of outside groups (i.e., the average resident) for one period of tenure. The way we calculated all variables accounted for variation across years, so we set Eqs. (1) and (2) equal to each other and solved for the length of a bachelor's reproductive lifespan, $L$ (Eq. (3)):

$$L = \frac{T(R_i + R_o N_r)}{R_b N_b}. \tag{3}$$
**Table 1 Resident parameters.**

| Statistic | Annual within-group siring rate in small groups ($R_i$) | Annual within-group siring rate in large groups ($R_i$) | Annual siring rate per neighboring group ($R_o$) | Number of neighboring groups ($N_r$) | Tenure length in years ($T$) |
|---|---|---|---|---|---|
| Mean ± SD | 1.4 ± 0.9 | 2.9 ± 1.7 | 0.1 ± 0.3 | 4.4 ± 1.5 | 2.8 ± 2.1 |
| Median | 1.3 | 3.1 | 0 | 4 | 2 |
| Range | 0.5–3 | 0–5 | 0–1 | 1–7 | 1–8 |
| N | 5 residents in 4 study groups | 6 residents in 4 study groups | 20 residents in the study pop, 22 periods of residency | 45 group-years | 21 residents in the study pop, 23 periods of residency |

The length of male reproductive lifespans varies widely among cercopithecines (4.3 years in Japanese macaques to 12.7 years in mandrills, *Clutton-Brock & Isvaran, 2007*). We do not know the length of male reproductive lifespans in blue monkeys, so we used estimates of female lifespan to identify plausible values. The oldest female in the study population was 33.5 years old (*Cords & Chowdhury, 2010*) and males attain full body size around 9 years old, so we estimated the maximum male reproductive lifespan to be the difference between these two values, 24.5 years. We considered it to be impossible for a bachelor to sire as many offspring as a resident if Eq. (3) yielded an estimate of $L$ that exceeded 24.5 years.

Given the large range in tenure length in our population (1–8 years) and the importance of tenure in the above equation, we repeated the calculations using the same parameters but replacing average tenure length with minimum or maximum tenure length. We also doubled all values of $L$ to compare the average bachelor to the average resident with 2 periods of tenure. These repetitions allowed us to compare bachelor and resident success in a total of 12 unique circumstances.

## Part II Results

### Resident parameters

The mean annual siring success in a resident's own group was 1.4 ± 0.9 offspring per year in small groups ($N = 5$ residents in 4 study groups) and 2.9 ± 1.7 offspring per year in large groups ($N = 6$ residents in 4 study groups; Table 2). Four residents sired offspring in adjacent groups: 3 residents sired 1 offspring and 1 sired 9 offspring over his 8-year tenure. Despite high outside-group siring success by some males, the average rate across residents was low (mean ± sd = 0.1 ± 0.3 offspring per adjacent group per year, $N = 22$ periods of residency; Table 1). The number of adjacent groups varied across group-years, but given low rates of outside-group siring success, reproduction in adjacent groups contributed much less to the total number of offspring sired during resident tenure than within-group siring success.

Tenure length varied greatly across residents (mean ± sd = 2.8 ± 2.1, range = 1–8 mating seasons, $N = 23$ periods of residency; Table 1), with observed values skewed towards shorter tenure lengths. Three males maintained residency for 6 years each, but only 1 was resident for the maximum 8 years.
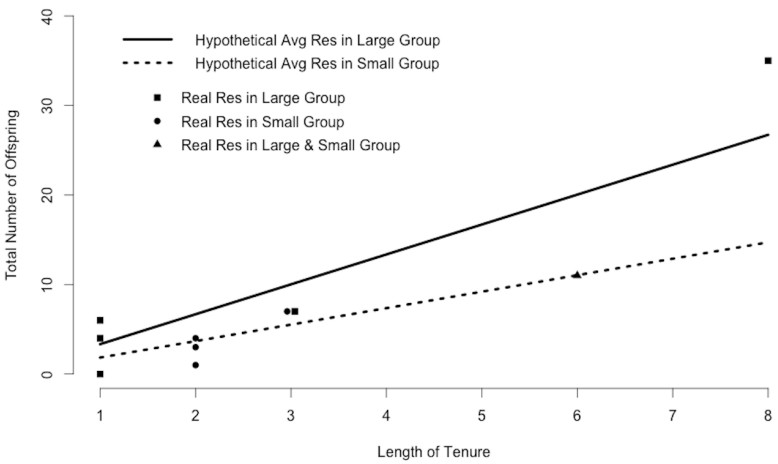

**Figure 3 Resident siring success.** Predicted siring success of hypothetical average residents as calculated by substituting mean values for $R_i$, $R_o$, and $N_r$ into Eq. (1). Points correspond to 10 real residents in our study population and indicate the total number of offspring sired in the study groups during one period of tenure. One male was resident in both a large and a small group.

**Table 2 Bachelor parameters.**

| Statistic | Annual siring rate per group ($R_b$) | Number of groups ($N_b$) |
|---|---|---|
| Mean ± sd | 0.1 ± 0.1 | 3.2 ± 0.4 |
| Median | 0 | 3.5 |
| Range | 0–0.3 | 2.5–3.5 |
| N | 29 bachelors | 5 bachelors |

Plugging mean values for resident within-group and outside-group siring success and the number of outside groups into Eq. (1) indicated that the average resident in a small group produces 1.4 offspring in his own group and 0.4 offspring in adjacent groups each year, for a total of 1.8 offspring annually. Maintaining this rate would yield 5.0 offspring during a period of tenure of average length and 14.7 offspring during a period of tenure of maximum length (Fig. 3). In contrast, the average resident in a large group produces 3.3 offspring annually, 9.2 offspring during a period of tenure of average length, and 26.8 offspring during a period of tenure of maximum length (Fig. 3). Many of the 10 residents in our study (excluding Ro, who seemed to be infertile; *Roberts, Nikitopoulos & Cords, 2014*) matched the siring profile of the hypothetical "average resident," siring approximately as many offspring as predicted by the calculations (Fig. 3).

### Bachelor parameters

An average bachelor sired 0.1 offspring per study group per year (Table 2). We found little variation in the number of group home ranges that a bachelor used (mean ± sd = 3.2 ± 0.4, range = 2.5–3.5 groups, Table 2). Using mean values for $R_b$ and $N_b$, the average bachelor would sire 0.3 offspring per year. There was some variation in siring

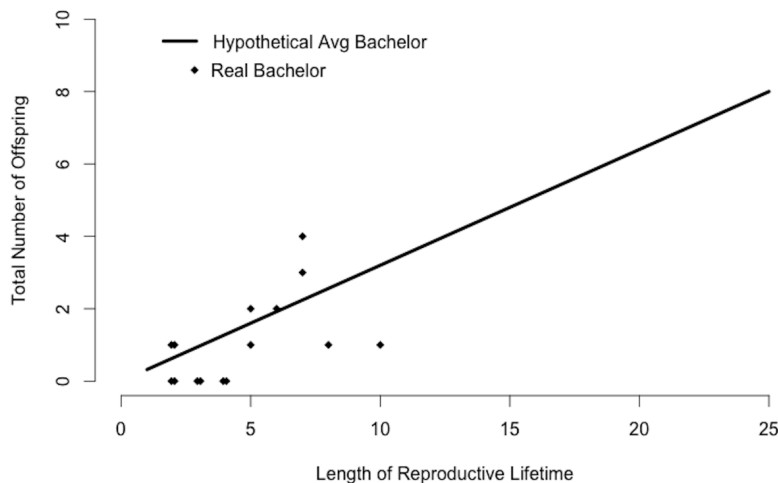

**Figure 4 Bachelor siring success.** Predicted siring success of a hypothetical average bachelor as calculated by substituting mean values for $R_b$ and $N_b$ into Eq. (2). Points correspond to 15 real bachelors and indicate the total number of offspring sired in the study groups during the years the bachelor was observed in our study population.

**Table 3 Comparison of resident and bachelor success.** The number of years ($L$) an average bachelor would need to pursue the bachelor tactic to sire as many offspring as an average resident with one (A) or two (B) periods of tenure of various lengths. Gray cells correspond to values that exceed a male's maximum reproductive lifespan (24.5 years) and therefore circumstances in which a bachelor could not sire as many offspring in his lifetime as a resident sires during his tenure.

|  | Min tenure (1 year) | | Mean tenure (2.8 years) | | Max tenure (8 years) | |
|---|---|---|---|---|---|---|
|  | Small group | Large group | Small group | Large group | Small group | Large group |
| **A. Average resident with one period of tenure** | | | | | | |
| **L for the average bachelor** | 5.8 | 10.4 | 16.1 | 29.3 | 46.0 | 83.5 |
| **B. Average resident with two periods of tenure** | | | | | | |
| **L for the average bachelor** | 11.6 | 20.8 | 32.2 | 58.6 | 92.0 | 167.0 |

success among 15 bachelors present in our study population for multiple years (Fig. 4). While some had no offspring in our study groups, others sired multiple offspring and several matched the siring profile of the hypothetical "average bachelor," predicted by the calculations (Fig. 4).

*Bachelorhood versus residency*

Our calculations indicate that the average bachelor may eventually sire as many offspring as the average resident who is present for one period of tenure of minimum length in either a small or large group (Table 3). It will always take less time for the average bachelor to match the siring success of a resident in a small group than a resident in a large group, so there are more circumstances that allow the average bachelor to sire as many offspring as the average resident in a small group. For example, the average bachelor may catch up to the average resident who is present for one period of tenure of average length if that resident is in a small group, but not if he is in a large group. The average bachelor would never be able to

sire as many offspring as a resident if that resident remains in his group for the maximum tenure length, regardless of group size (Table 3).

Doubling all values of $L$ indicates that there are fewer circumstances in which the average bachelor can catch up with the average resident with two periods of residency. Specifically, a bachelor may be as successful as a resident with two periods of residency only if both periods of tenure are short (Table 3). If the average resident has one period of tenure in a small group that lasts the minimum tenure length and a second period of tenure in another small group that lasts that mean length, a bachelor may match the resident's siring success in 21.9 years, which is less than our estimate of the maximum male reproductive lifespan.

# DISCUSSION

## Factors affecting extra-group male siring success

The relationship between rank and male mating and siring success has been a topic of investigation for decades (e.g. *Cowlishaw & Dunbar, 1991*; *Ellis, 1995*; *Majolo et al., 2012*). Studies have focused on species living in multi-male groups where males interact regularly and competitors are in close proximity. To our knowledge, our study is the first to rank extra-group males and use those ranks to predict siring success in a one-male species. Although the resident male of a group dominates within-group reproduction in blue monkeys (*Roberts, Nikitopoulos & Cords, 2014*), other males also participate, and do so differentially. Our results suggest that higher-ranking extra-group males have a substantial siring advantage; the odds of siring an infant by increased roughly by a factor of 156 with a one-step increase in ordinal rank. High-ranking males include the residents of neighboring groups, which were represented disproportionately among the extra-group sires. Neighboring residents were present during the conceptive period of nine infants and sired six of these infants.

Among bachelors, rank was not a significant predictor of siring success, which suggests that bachelors do not engage in contests with other bachelors for access to reproductive opportunities. Indeed, the majority of observed male–male contests involve at least one resident (M Cords & SJ Roberts, pers. obs., 2014). Bachelor blue monkeys probably use a highly opportunistic tactic to reproduce, mating when they encounter a sexually active female rather than queuing for reproduction as seen in wild savannah baboons (*Alberts, Watts & Altmann, 2003*). Opportunistic matings may be more likely to occur in forest-dwelling taxa than in those inhabiting open habitats, because limited visibility decreases the chance that a higher-ranking competitor would interfere (*Rowell, 1988*). We have seen bachelors copulate with females in our study population near to but on the other side of vine tangles from a resident, suggesting that limited visibility does allow bachelors to steal copulations. The dense forest habitat may similarly negate any effect of dominance rank on siring success among bachelors, allowing low-ranking bachelors to mate and reproduce opportunistically, despite the presence of other higher-ranking bachelors nearby.

It is perhaps more surprising that the time an extra-group male visited the group during a mother's conceptive period did not significantly predict siring success. Even in the

absence of contests, one might expect extra-group males that are more consistently present to sire more offspring. Perhaps, however, rare opportunities to mate with conceptive females (copulations and mounts during conceptive periods occurred at a rate of 0.21 events/hour; unpublished data from 86 h of focal samples of 48 adult females in their conceptive period) combined with the difficulty of monitoring their sexual activity in large and widely dispersed groups (group spread: mean ± sd = 81 ± 46 m, range = 15–282 m, unpublished data from 198 measurements of 6 groups with 8–49 individuals) in a visually opaque environment introduces a large element of serendipity into relative mating (and ultimately reproductive) success of bachelor males.

## Comparing male reproductive tactics

If bachelor male blue monkeys are pursuing an alternative reproductive tactic instead of simply making the best of a bad job, we would expect the lifetime reproductive success of a lifelong bachelor to resemble that of a male who incorporated a period of residency into his reproductive lifespan. Although resident males have a verified reproductive advantage in the short term and on a small scale (*Roberts, Nikitopoulos & Cords, 2014*), a bachelor may be able to make up for this reproductive disadvantage by reproducing for a longer period or in more groups (*MacLeod, Ross & Lawes, 2002*). Our calculations, however, indicate that it will usually take many years for a bachelor to sire as many offspring as a resident during one period of tenure, and in most circumstances, the resident tactic probably results in higher lifetime reproductive success than the bachelor tactic.

Although residency usually confers a reproductive advantage, we identified some circumstances in which an average bachelor's reproductive success would match an average resident with one or two periods of tenure. Specifically, an average bachelor would take 5.8 years to match the siring success of an average resident in a small group with a period of tenure of minimum length. This estimate increases to 10.4 years when one compares an average bachelor to an average resident with minimal tenure in a large group and to 16.1 years in comparison to the average resident in a small group with period of tenure of average length. There were fewer circumstances in which the average bachelor could sire as many offspring as the average resident with two periods of tenure, but it was not impossible.

When calculating resident and bachelor parameters, we made several decisions that affected the comparison. We limited resident siring success to the period of residency because the available genetic and behavioral evidence suggests that resident males rarely sire offspring before or after becoming resident. Our data indicate that reproduction by males with at least one period of residency occurred almost entirely during their residency, but it remains possible that this restriction caused us to underestimate resident lifetime reproductive success. Specifically, if residents do sire offspring before and after residency *and* they both sire offspring at the same rate and live as long as life-long bachelors, the lifetime reproductive success of a resident will always be greater than the lifetime reproductive success of a bachelor.

When calculating annual siring success for bachelors, we eliminated young bachelors who had recently left their natal groups and temporary visitors that were unable to be recognized. These bachelors had very low siring success, so omitting them from our calculation increased the average bachelor siring success and gave the average bachelor a better chance of catching up with a resident. If we included these males, the rate of bachelor siring success was half of the estimated values (0.05 offspring per group per year instead of 0.1 offspring per group per year), which would double the number of years required for the average bachelor to sire as many offspring as a resident. The average bachelor would be unable to catch up with the average resident with one period of tenure unless the resident had a very short tenure. The average bachelor would be able to catch up with the resident with two periods of tenure only if the resident had two very short periods of tenure in a small group.

We judged the estimates of bachelor reproductive lifespan to be plausible if they were less than 24.5 years, based on the difference between maximum female lifespan and the age at which males attain full adult body size. This male reproductive lifespan represents a maximum, as it is unlikely that male blue monkeys live as long as females. In long-lived species that live in one-male or multi-male groups, males typically show higher annual mortality rates and a more rapid decline in survival with age (*Clutton-Brock & Isvaran, 2007*; *Bronikowski et al., 2011*). If we estimated male reproductive lifespan more conservatively by assuming that the oldest reproductive male was as old as the oldest *reproductive* female (26.5 years), the male reproductive lifespan would be 17.5 years instead of 24.5. This change means there would be fewer circumstances in which a bachelor male could sire as many offspring as a resident during one period of tenure, but there would still be some (Table 3).

## CONCLUSION

Although resident male blue monkeys sire most offspring born in their groups, they lose about 40% of paternity to outside males (*Roberts, Nikitopoulos & Cords, 2014*). When a resident male did not monopolize reproduction in his group, the rank of extra-group males influenced their relative success, with high-ranking males (who were often neighboring residents) having a special advantage. Neither bachelor rank nor male presence affected siring success relative to other bachelors, however, suggesting that bachelors do not use contest competition to allocate reproductive opportunities and that opportunistic matings play an important role in bachelor siring success.

Bachelor males sire offspring at a much lower rate than do residents, but a bachelor may be able to make up for this reproductive disadvantage by reproducing for a longer period of time or in multiple groups. Our comparison of average bachelor and resident siring success indicated that in most circumstances, a lifelong bachelor would be unlikely to sire as many offspring during his lifetime as a resident during one or two periods of residency. However, a bachelor who sires offspring at the average rate in multiple groups for several years may have similar lifetime reproductive success as a male whose reproduction is limited to one average period of residency, especially in a small group. Our

findings thus suggest that one should not assume a resident reproductive advantage for males in one-male groups in all circumstances.

## ACKNOWLEDGEMENTS

We thank the Government of Kenya for permission to conduct the research, and the Centre for Kakamega Tropical Forest Studies at Masinde Muliro University of Science and Technology for local sponsorship. Genetic resources utilized in this study were provided by the government and people of Kenya. We are grateful to the local forest station personnel, and many field and lab assistants. We especially thank T. Disotell for hosting the genetic analysis and D. Rabinowitz for statistical advice.

### Funding

This work was supported by the National Science Foundation (grants BCS05-54747, BCS10-28471, a Graduate Research Fellowship) and the LSB Leakey Foundation. The funders had no role in study design, data collection and analysis, decision to publish, or preparation of the manuscript.

### Grant Disclosures

The following grant information was disclosed by the authors:
National Science Foundation: BCS05-54747, BCS10-28471.
LSB Leakey Foundation.

### Competing Interests

Su-Jen Roberts is an employee of New Knowledge Organization Ltd. and Su-Jen Roberts and Marina Cords are members of New York Consortium in Evolutionary Primatology.

### Author Contributions

- Su-Jen Roberts conceived and designed the experiments, performed the experiments, analyzed the data, contributed reagents/materials/analysis tools, wrote the paper, prepared figures and/or tables, reviewed drafts of the paper.
- Marina Cords conceived and designed the experiments, performed the experiments, analyzed the data, contributed reagents/materials/analysis tools, wrote the paper, reviewed drafts of the paper.

### Animal Ethics

The following information was supplied relating to ethical approvals (i.e., approving body and any reference numbers):

Methods were approved by the Columbia University IACUC (#AC-AAAD9003), the National Council for Science and Technology, the Kenya Wildlife Service and National Environmental Management Authority.

## Field Study Permissions

The following information was supplied relating to field study approvals (i.e., approving body and any reference numbers):

Methods were approved by the Columbia University IACUC (#AC-AAAD9003), the National Council for Science and Technology, the Kenya Wildlife Service and National Environmental Management Authority.

## Supplemental Information

Supplemental information for this article can be found online at http://dx.doi.org/10.7717/peerj.1043#supplemental-information.

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
