# Peer review of "Life as a bachelor: quantifying the success of an alternative reproductive tactic in male blue monkeys"

_PeerJ, doi:10.7717/peerj.1043_

## Round 0.1 · original submission · Minor Revisions

Please revise, paying particular attention to the comments of the more critical reviewer.

·

Basic reporting

This manuscript is a highly significant contribution to our knowledge of primate behavioral and reproductive ecology, especially with regard to cercopithecine monkeys living in single male/multiple female groups.

Years ago, the eminent primatologist Thelma Rowell wrote a paper concerning sexual dimorphism among polygynous primates in which she emphasized the critical and ill-understood role of competition between bachelor males. This long-term study by Roberts & Cords about blue monkeys in Kakamega forest adds important new knowledge concerning group dynamics not only among adults that are invested within the mating system but also among the bachelor males.

Experimental design

everything concerning the experimental design is thorough and excellent

Validity of the findings

There is no doubt that the findings are completely valid.

Additional comments

As a person who mainly studies fossil monkeys, its great to see such stellar research which can potentially be very useful for primate paleontologists in attempting to infer the mating systems of ancestral primates.

Reviewer 2 ·

Basic reporting

No comment

Experimental design

No comment

Validity of the findings

See below

Additional comments

This manuscript exams factors contributing to extra-group male reproductive success and attempts to model the fitness consequences of being a group leader or bachelor male. The first portion of the paper – the analysis of bachelor reproductive success is very interesting and impressive, especially given the challenges of identifying and tracking bachelor males. The study certainly shows the value of long-term field studies of animal behavior. About this part of the paper I have only some minor comments and suggestions.

Ln 192. It says that success increased by a factor of 1.01 for each Elo rating. I take it that this means that for each point in the Elo ranking a male’s chance of success increases by 1%. Is that correct. The way it is written it is somewhat difficult to interpret and I would suggest that authors try to find more plain language to describe the effect of rank on success. The fact that rank matters is important and sets the stage for thinking about the traits in males that make them more competitive and thus are likely to be under selection. To me, this seemed like an important finding that doesn’t really stand out in the results.

A general comment about this section is that it might be useful to provide a table or chart of some sort that shows that number of bachelor males seen in each group over some period of time throughout the study (per month? per quarter? ) to give the readers a clearer sense of what the dynamics are. The that end, a similar chart/table from the perspective of bachelors may be interesting as well – for any given bachelor how often do he face competition with other bachelors, how many groups does he visit in a given breeding season? Etc.


The second section of the paper – the modeling of fitness outcomes for bachelor versus resident males – I found more problematic. The approach used here assumes that resident male and bachelor males are distinct tactics despite the fact that the authors repeatedly mention that resident males are bachelors both before and after residency. While the authors observed that 9 of 10 males that were seen as bachelors and residents did not reproduce as bachelors, it seems like a logical leap to then conclude that the siring success of a male who attains residency is dependent on the residency period. Are they any worse at attaining bachelor matings than bachelors? Are they dying when they are deposed? Treating resident males and committed bachelors as distinct entities to model lifetime fitness would only make sense if they were distinct strategies, which I am not convinced that they are.

In the paper, the authors appear to conclude that resident reproductive success is determined by his tenure as a resident, with little success as a bachelor. Is it really the case that males that become residents have lower success as bachelors compared to committed bachelors? If that is the case, is it due to other factors such as age (which I appreciate the authors will not know for many if not most bachelors). For example, one could imagine that all young males have low siring success and are generally less competitive. At some peak age, some males are able to acquire groups while others stay as bachelors, but are among the most competitive bachelors. As males age, they may senesce and become less competitive, loosing their residency status and returning to the bachelor pool. Under such a scenario we might appear that resident males are poor competitors as bachelors, when in fact they are merely young or old. Additionally, such a model highlights why it is likely to be unreasonable to consider bachelor success as a steady rate of reproduction throughout a male’s adulthood.

Another piece of evidence that could provide some insight into whether or not committed bachelors are a different strategy than resident males would be if there were differences in longevity. It is plausible that residency could shorten a male’s lifespan, especially if deposed residents tended to be wounded or killed in take-overs. From the fact that the authors mention that some residents return to the bachelor pool, this seems not to be the case. In any event, male life history in a species with male dispersal is very hard to measure and not something that I would expect the authors to be able to address directly, but perhaps it is worth thinking about.

---

## Round 0.2 · accepted · Accept

Thank you for your conscientious attention to detail in your revisions.